# MCP1 Could Mediate FGF23 and Omega 6/Omega 3 Correlation Inversion in CKD

**DOI:** 10.3390/jcm11237099

**Published:** 2022-11-30

**Authors:** Deborah Mattinzoli, Stefano Turolo, Carlo Maria Alfieri, Masami Ikehata, Lara Caldiroli, Silvia Armelloni, Giovanni Montini, Carlo Agostoni, Piergiorgio Messa, Simone Vettoretti, Giuseppe Castellano

**Affiliations:** 1Renal Research Laboratory, Fondazione IRCCS Ca’ Granda Ospedale Maggiore Policlinico, 20122 Milan, Italy; 2Pediatric Nephrology, Dialysis and Transplant Unit, Fondazione IRCCS Ca’ Granda Ospedale Maggiore Policlinico, 20122 Milan, Italy; 3Department of Nephrology, Dialysis and Renal Transplantation, Fondazione IRCCS Ca’ Granda Ospedale Maggiore Policlinico, 20122 Milan, Italy; 4Department of Clinical Sciences and Community Health, University of Milan, 20122 Milan, Italy; 5Pediatric Intermediate Care Unit, Fondazione IRCCS Cà Granda Ospedale Maggiore Policlinico, 20122 Milan, Italy

**Keywords:** CKD, fatty acids, FGF23, inflammation

## Abstract

Fibroblast growth factor 23 (FGF23) concentrations rise after the early stages of chronic kidney disease (CKD). FGF23 is involved in inflammatory reactions closely associated with an incremented risk of cardiovascular disease (CVD). There is growing evidence that omega-6 (n-6) and n-3 polyunsaturated fatty acids (PUFA) can modulate inflammation through several mediators producing an opposite effect on cardiovascular (CV) risks. In this study, we explore whether there is any correlation between PUFA, FGF23, and inflammation in CKD patients. We evaluated, cross-sectionally, 56 patients at different stages of CKD. Monocyte chemoattractant protein 1 (MCP1), and intact and c-terminal FGF23 (iFGF23, cFGF23) were quantified by the ELISA, and the fatty acids (FA) profile was analyzed by gas chromatography. Concurrently with an eGFR decrease (*p* < 0.01) and an MCP1 increase (*p* = 0.031), we observed an inversion of the correlation between FGF23 and the n-6/n-3 ratio. This last correlation was inversed in CKD stage 3 (r^2^ (−) 0.502 *p* = 0.029) and direct in stage 5 (r^2^ 0.657 *p* = 0.020). The increase in MCP1 seems to trigger events in the inversion of the correlation between FGF23 and the n-6/n-3 PUFA ratio. This result strongly encourages future studies on basal pathways, on possible pharmacological interventions, and on managing kidney transplant patients treated with immunosuppressive therapy.

## 1. Introduction

Chronic kidney disease (CKD) affects almost 10% of the world’s population, and its prevalence is constantly increasing worldwide. More than 60% of patients in the earlier stages of CKD will never reach end-stage renal disease because of their increased cardiovascular (CV) risk [1,2]. The CV events’ prevalence appears early, already significantly high at CKD stage 3, with an elevated risk of death at CKD stages 4–5 [3].

CKD is accompanied from the early stage by a high level of fibroblast growth factor 23 (FGF23) as an adaptive mechanism to counterbalance the alteration of mineral metabolism [4]. At the renal level, FGF23 regulates phosphorus excretion and reduces the synthesis of 1,25(OH)_2_D_3_, and at the parathyroid gland level, FGF23 downregulates the synthesis of the parathyroid hormone [5]. Despite its compensative role, an impact of FGF23 in CV events has recently been demonstrated both in the general population and in CKD using extrarenal actions: (a) the stimulation of liver inflammatory cytokines production, (b) the induction of cardiac myocytes hypertrophy, (c) the harmful modulation of the nitric oxide/oxygen-free radicals ratio causing endothelial dysfunction, atherosclerosis, and vascular calcification [6,7,8]. In summary, FGF23 is considered an excellent early CKD biomarker and an active mediator of cardiovascular disease (CVD), so new interventions to lower its effects are eagerly researched [9].

CKD is characterized by a change in serum fatty acid (FA) levels, which contributes to pro-inflammatory, pro-atherogenic, and oxidative stress, all of which have a negative impact on the heart, accelerating CKD progression and increasing CVD risk [10]. The role of polyunsaturated fatty acids (PUFA) in CVD risk reflects their action to support cell structure and homeostasis, mainly via their metabolites [11]. Thus, a change in PUFA concentrations due to CKD could generate an imbalance in their components, which become signaling mediators and correlate with clinical outcomes in several chronic diseases [12]. There is growing evidence that n-3 and n-6 PUFAs can positively and negatively modulate the inflammatory response, influencing CV risk in an opposite and contrasting way [13,14]. The n-3 PUFA improves the circulating lipid profile by an inverse relationship with the triglyceride level, a positive correlation with high-density lipoprotein, and a modification of the low-density protein composition [15,16]. Their membrane incorporation instead of n-6 PUFA induces a change in membrane fluidity/elasticity, increasing cell communication and positively affecting vascular endothelial and smooth muscle cells [17,18]. The n-3 eicosapentaenoic acid (EPA) increases nitric oxide release leading to vascular relaxation and prevents the transformation of n-6 arachidonic acid into pro-inflammatory eicosanoids causing platelet aggregation and vasoconstriction [19,20,21].

It was observed that the oxidative stress typical of CKD patients causes a loss of n-3 biological function, increasing CVD risk, and claiming new strategies [22,23,24].

The monocyte chemoattractant protein-1 (MCP1) is an essential promoter of inflammation that stimulates and attracts circulating monocytes to inflamed tissue. Then, their differentiation into inflammatory cytokines in the kidney exacerbates proteinuria, tubulointerstitial fibrosis, and glomerulosclerosis [25]. On the endothelial cells, the monocytes recruitment by MCP1 suggest its crucial role in the development of the atherosclerotic lesion leading to severe cardiac consequences [26]. In bone, MCP1 is a well-recognized stimulator of bone resorption leading to an increase in FGF23. MCP1 recovers a central role in CKD, CVD, and mineral bone disorder (MBD) [27].

Therefore, this study explores the possible correlation between i/cFGF23, PUFA, and MCP1 in CKD patients (stages 3, 4, and 5 ND).

## 2. Materials and Methods

### 2.1. Patients

We evaluated 56 patients > 18 years old attending the outpatient nephrology clinic at our institution. To be included, they had to have an estimated glomerular filtration rate (eGFR) < 60 mL/min/1.73 m2 and not in dialysis. The eGFR was obtained using CKD-EPI formula. The cohort was divided in 3 groups according to their CKD stage: stage 3 (30 < eGFR < 59 mL/min); stage 4 (15 < eGFR < 29 mL/min) and stage 5 ND (eGFR < 15 mL/min). Exclusion criteria were active cancer, symptomatic infectious disease in the previous two months, decompensated chronic liver diseases, symptomatic heart failure (NYHA II–IV), endocrine disease different to mineral metabolism anomalies, intestinal malabsorption, hospitalization in the last two months, and inability to cooperate. We also excluded all patients under treatment with immunosuppressive drugs and with a presumed overall life expectancy of <6 months. The participant’s blood samples were collected in the morning on the same day of the visit after overnight fasting for at least eight hours. In addition, the 24 h urine samples were collected for routine analysis (eGFR and phosphaturia). 

Ethical approval: The study was conducted according to the ICP Good Clinical Practices Guidelines and to the declaration of Helsinki, and the approval of the Ethics Committee of our institution (approval document 347/2010, PROVE: Proteinuria and Vascular Endpoints). All patients signed informed consent to participate in the study, as specified in the ICMJE recommendations.

### 2.2. Fatty Acid Analysis

A serum aliquot of 50 µL was collected during a routine check. The samples were transferred into vials and methylated with 800 µL of hydrochloric acid solution 3N in methanol (HClMe 3N) (Sigma-Aldrich, St. Louis, MO, USA) and incubated for 1 h at 90 °C. Then, the sample was refrigerated at 4 °C for 10 min. Afterward, 2 mL of potassium chloride (KCl) solution and 330 µL hexane (Sigma-Aldrich) were added. Samples were first vortexed and then centrifuged at 3000 rpm for 10 min. Finally, the hexane layer (the upper layer) was collected from each vial and transferred into a gas chromatography vial for FA profile evaluation with gas chromatographer Shimadzu Nexis GC-2030 (Shimadzu, Kyoto, Japan) equipped with a 30 m fused silica capillary column FAMEWAX Restek (Restek, Bellefonte, PA, USA). The gas chromatography results were analyzed using Lab solution software 5.97 SP1 (Shimadzu, Japan). Both single and fatty acid groups (PUFA, PUFA n-3, PUFA n-6) are expressed as relative percentages of total considered fatty acids, whose value is always 100. The urinary fatty acids were not analyzed since the excreted one cannot influence the pathways related to FGF23 and MCP1 in the blood.

### 2.3. Enzyme-Linked Immunosorbent (ELISA) of FGF23 Intact/C-Terminal and Monocyte Chemoattractant Protein 1 (MCP1) 

Plasma FGF23 intact and c-terminal (iFGF23 and cFGF23) were measured by a second-generation two-site enzyme-linked immunosorbent assay ELISA Kit (Immutopics Quidel Co., San Diego, CA, USA) as previous studies [28,29]. The minimal detectable concentration is 1.5 pg/mL and 1.5 RU/mL respectively. The coefficient of variation was: (i) iFGF23: intra-assay 4.1% and 2%, at 43 and 426 pg/mL and inter-assay 9.1% and 3.5%, at 46 and 441 pg/mL, (ii) cFGF23: intra-assay 2.4% and 1.42% at 33.7 and 302 RU/mL, inter-assay 1.4%. at 33.6 and 293 RU/mL, respectively. Absorbance in each well was read at a dual wavelength of 450/630 nm.

Serum MCP1 level, considered as marker of inflammation, was evaluated by the commercially available ELISA Kit (R&D Systems, Inc., Minneapolis, MN, USA) [30,31]. The minimal detectable concentration was 1.7 pg/mL. The coefficient of variation for the intra-assay was 7.8% and 4.7% at 76.7 and 364 pg/mL, and for the inter-assay, 6.7% and 5.8% at 74.2 and 352 pg/mL, respectively. Absorbance in each well was read at a dual wavelength of 450/570 nm. For both ELISA, the replicate background measurements were subtracted from all 450 nm measures. 

### 2.4. Statistical Analysis

The data were correlated by two-tailed Spearman bivariate analysis. Negative correlations are expressed with the prefix “(-)”. The bivariate correlation graph also shows 95% regression. The Kruskal–Wallis test was used to assess differences between CKD groups; the graphs also report the total median line for each considered parameter.

All descriptive tables report data as mean and standard deviation unless otherwise specified. With t-test analysis, differences among demographic and clinical biomarkers were assessed; the chi-square test was used to statistically assess differences for gender, diabetics, and cardiovascular events among CKD groups. A *p* < 0.05 was considered statistically significant for all the statistical analyses. All the analyses were performed with SPSS21 (IBM, Armonk, NY, USA) software.

## 3. Results

Demographic and biochemical data of the considered cohort are reported in Table 1. The average age was 78 ± 8 years old; 66% were male, 59% were diabetics, and 55% had previous CV events. Patients were divided into three groups according to their CKD stage (see method): 19 patients to stage 3, 25 patients to stage 4, and 12 patients to stage 5.

The linear trend in biomarkers eGFR, MCP1, and i/cFGF23 were evaluated during each CKD stage (Figure 1A–D). Kruskal–Wallis’s analysis showed that during the eGFR decline (*p* < 0.01 from median 23.5 mL/min), we observed an increase in MCP1 (*p* = 0.031 from median 470.4 pg/mL), iFGF23 (*p*-value <0.01 from median 91.5 pg/mL), and cFGF23 (*p* = 0.03 from median 140 RU/mL) from CKD stage 3 to stage 5.

Then the PUFA profile among CKD stages was analyzed and reported in Table 2. An increase in total PUFA was observed during the renal function decline from CKD 4 to 5 (*p* = 0.009). Exploring the PUFA n-3, a significant decrease in n-3 from CKD 3 to 4 (*p* = 0.03) was observed, and in detail, a reduction in n-3 docosapentaenoic acid (DPA) (22:5) from CKD 3 to 4 (*p* = 0.04).

Conversely, regarding n-6, an increase was observed from CKD 3 to 5 (*p* = 0.03) and CKD 4 to 5 (*p* = 0.01), and in detail, an increase in n-6 Linoleic acid (18:2) from CKD 3 to 5 (*p* = 0.03) and from CKD 4 to 5 (*p* = 0.006) (Table 2).

Therefore, analyzing the n-6/n-3 ratio, which helps interpret the PUFA results, we observed an increasing trend among the CKD stages (*p* = 0.03 from median 12.43) (Figure 2).

To better explore the connection between PUFA and i/cFGF23, we analyzed their correlation in the overall cohort, and, interestingly, no correlation was found between PUFA and i/cFGF23. For this reason, only the correlation found among the different CKD stages are reported.

In CKD stage 3, a positive correlation was observed between n-3 and iFGF23 (r^2^ 0.456, *p* = 0.050), which in turn correlated negatively with the ratio n-6/n-3 (r^2^ (−) 0.502, *p* = 0.029). Notably, even if not significantly, cFGF23 followed the same behaviour (Figure 3A).

Conversely, in CKD stage 5, the trend was reversed, and a negative correlation appeared between cFGF23 (r^2^ (−) 0.587, *p* = 0.045) and n-3, and a positive one appeared with n-6/n-3 (r^2^ 0.657, *p* = 0.020). Additionally, in this case, iFGF23 followed the same behaviour non-significantly (Figure 3B).

In detail, in CKD stage 3, iFGF23 correlated positively with n-3 DPA 22:5 (r^2^ 0.628, *p* = 0.004), whereas in CKD5, cFGF23 correlated positively with the n-6 Osbond acid 22:5 (r^2^ 0.632, *p* = 0.028) and negatively with n-3 docosahexaenoic acid (DHA) 22:6 (r^2^ (−) 0.627, *p* = 0.029) (Appendix A). Notably, no correlation was observed during CKD stage 4 (Appendix A).

We then observed the correlation first between PUFA and the inflammatory marker MCP1 and then with i/cFGF23 among the CKD stages.

Starting from MCP1, no correlation appeared, either with n-3 or with n-6 during CKD stage 3.

During CKD stage 4, while no correlation was observed between MCP1 and n-3, a positive one appeared between n-6 and MCP1. In particular, in CKD stage 4, MCP1 correlated with n-6 dihomo-γ-linolenic acid (DGLA) 20:3n6 (r^2^ 0.594 *p* = 0.002) and n-6 Osbond acid 22:5n6 (r^2^ 0.521 *p* = 0.009), and above all with the major contributors to the production of inflammatory kidney mediators n-6 arachidonic acid 20:4n6 (r^2^ 0.424 *p* = 0.039). In CKD5, MCP1 correlated with the precursor of arachidonic acid, namely n-6 linoleic acid 18:3n6 (r^2^ 0.677 *p* = 0.016), and no correlation continued with n-3 (Table 3).

We then analyzed the correlation between iFGF23 and cFGF23, and MCP1; a positive one occurred only with cFGF23 (r^2^ 0.04 *p* = 0.28) (Figure 4).

## 4. Discussion

The ongoing interrelation between the renal/extrarenal action of FGF23 and the inflammation in CKD gives place to a cascade event strongly associated with CVD leading us to a broad range of study approaches. Considering that the derangement of FA metabolism characterizes CKD due to chronic inflammation, oxidative stress, and malnutrition accelerating CKD and CVD progression, in the present study, we investigate if a correlation among FGF23, MCP1, and PUFA occurs.

In the first step of our study, we observed a not surprising progressive increase in both i/cFGF23 and pro-inflammatory factor MCP1 during the renal function decline in our ND cohort. The inflammation stimulates FGF23 through several mechanisms, activating bone resorption and increasing phosphate and calcium that stimulates FGF23 production [32,33]. The MCP1 itself is a recruiter of osteoclast precursors, increasing the bone resorption and then the FGF23 levels in addition to being a great contributor to the worsening renal function [34]. On the other hand, as already reported in our previous experiments, FGF23 stimulates the liver production of inflammatory cytokines [6,35].

Analyzing our cohort, during the renal function decline, an antithetic attitude manifested by an increase in PUFA n-6 and a decrease in PUFA n-3 appears. The increase in the arachidonic acid, the precursor n-6 linoleic acid (18:2), in our cohort strongly suggests its possible conversion in eicosanoids, prostaglandin, and leukotriene, all well-recognized activators of several pathological processes [36]. Moreover, the simultaneous decrease in n-3 DPA (22:5), usually involved in the production of solid attenuators of inflammation (protectins and resolvins), confirms the hypothesis of an ongoing inflammatory mechanism leading to the worsening of renal function [37].

Then, hypothesizing a possible FGF23 role in this lipid alteration possibly mediated by the inflammation, we analyzed first the FGF23 correlation with PUFA and then with MCP1. No significant correlation appeared between FGF23 and PUFA in the overall cohort, so this possible correlation was tested among the single CKD stages.

To facilitate the understanding and assessment of the PUFA data, several studies consider the n-6/n-3 ratio, whose growth is a sign of several diseases (CVD, cancer, and inflammatory/autoimmune diseases) [38,39]. While in CKD stage 3, we observed a negative correlation between the n-6/n-3 ratio and iFGF23, conversely, in CKD stage 5, we observed a positive correlation between the n-6/n-3 ratio and cFGF23. The last positive correlation occurred with the n-6 Osbond acid 22:5, a product of the arachidonic acid cascade, validating our previous hypothesis on the ongoing inflammatory process.

Particular attention should be given to the shift between the iFGF23 and cFGF23 correlation results with the n-6/n-3 ratio. Indeed, the FGF23 protein circulates as a full-length iFGF23 or cleaved shorter cFGF23 based on whether a fine post-transcriptional regulation occurs, influencing the level of the two forms [40]. David’s group study demonstrated that the induction of acute inflammation increases the FGF23 furin-like protease cleavage. Then, not surprisingly, in CKD stage 5, we observe the correlation of the n-6/n-3 ratio only with cFGF23 [41].

In CKD stage 4, the absence of any correlation between the i/c FGF23 and n-6/n-3 ratio indicates an apparent “state of transit”. However, CKD stage 4 is not a sleeping step because it is precisely in this step that the positive correlation between the pro-inflammatory protein MCP1 and PUFA n-6 begins. This last result suggests that MCP1 may be one of the agents causing the correlation inversion between the FGF23 and n-6/n-3 ratio observed in CKD stages 3 and 5.

As well as the already acclaimed increase in n-3 dietary/pharmacological intake, the present study suggests that a strategically more targeted approach is necessary. An interesting study by the Gosling study group reported a reduction in atherosclerosis in the transgenic mice MCP1 −/− and in the MCP1 receptor −/− [42,43]. According to our results, using MCP1 antagonists, already reported in experimental trials, might represent a potential future strategy for modulating the detrimental effects of FGF23 in CKD patients [44].

According to our knowledge, this is the first study exploring the interrelation between PUFA, FGF23, and inflammation among the separated stages of renal decline. A possible limitation of the study is the small degree of the cohort, but, on the other hand, the strict inclusion and exclusion criteria adopted to limit the potential confounding bias (excluding subjects with immunosuppressive therapies or already known to be affected by increased inflammatory responses) permit good homogeneity of the patients.

In conclusion, during the progression of renal disease, we observed an inversion of the correlation between the n-6/n-3 ratio and i/cFGF23, positive in CKD stage 3 and negative at stage 5. The increase in MCP1 serum concentration, along with the decline in renal function and its positive correlation with the most n-6 PUFA in CKD stage 4, strongly suggest this as being one of the possible responsible causes. The n-6/n-3 ratio is reconfirmed as a potential biomarker of increased CV risk strongly associated with FGF23 levels and its bidirectional connection with inflammation resulting in a higher risk for mortality in CKD for CVD. An increase in n-3 dietary/pharmacological intake, a reduction in systemic oxidative mechanisms affecting the n-3 structure, and possibly the use of MCP1 antagonists or agents interfering with its pathway could represent a new strategy for mitigating the detrimental effects of FGF23 in CKD patients. It could be interesting to extend this research to other CKD settings, such as dialysis and renal transplantation.

## Figures and Tables

**Figure 1 jcm-11-07099-f001:**
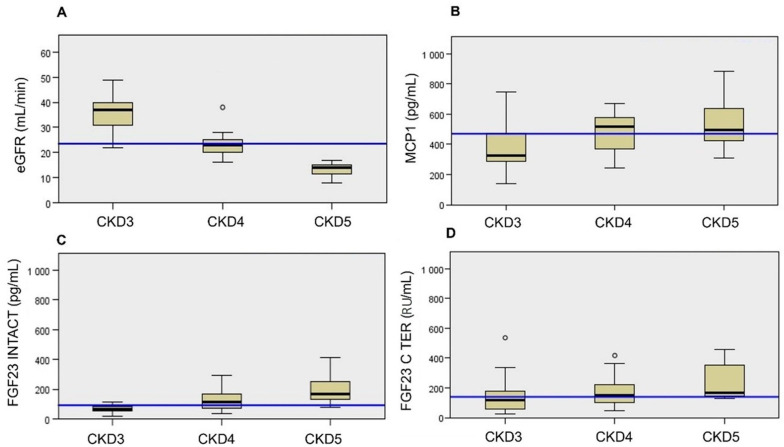
eGFR (**A**), MCP1 (**B**), i/cFGF23 (**C**,**D**) data according to CKD stage, blue line = median, *n* = 56 (*n* = 19 in CKD3, *n* = 25 in CKD4, *n* = 12 in CKD5). ° = outliner patients.

**Figure 2 jcm-11-07099-f002:**
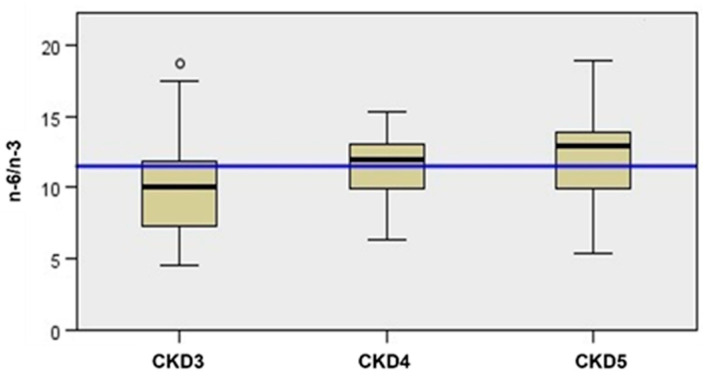
The n-6/n-3 trend data according to CKD stage, blue line = median, *n* = 56, (*n* = 19 in CKD3, *n* = 25 in CKD4, *n* = 12 in CKD5). Kruskal–Wallis’s analysis from median. ° = outliner patients.

**Figure 3 jcm-11-07099-f003:**
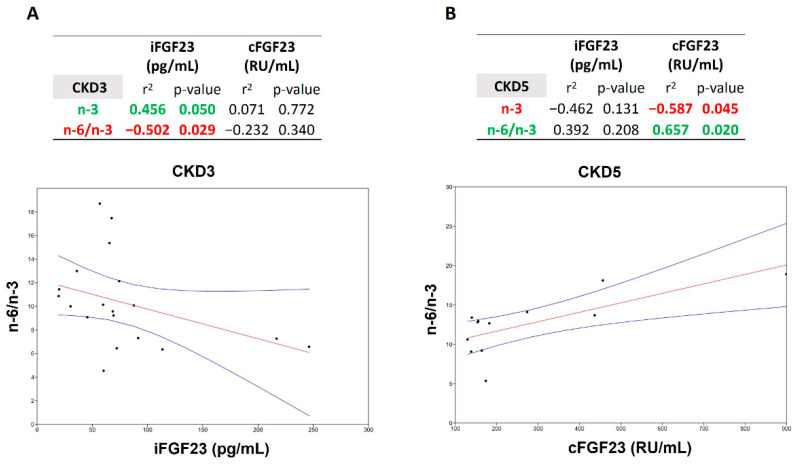
Correlation between n-6/n-3 and n-3 and i/cFGF23 among CKD stage 3 (**A**) and 5 (**B**). Significant positive correlation in green and negative in red. “−” indicates a negative correlation. Two-tailed Spearman bivariate analysis. *n* = 19 in CKD3, *n* = 12 in CKD5.

**Figure 4 jcm-11-07099-f004:**
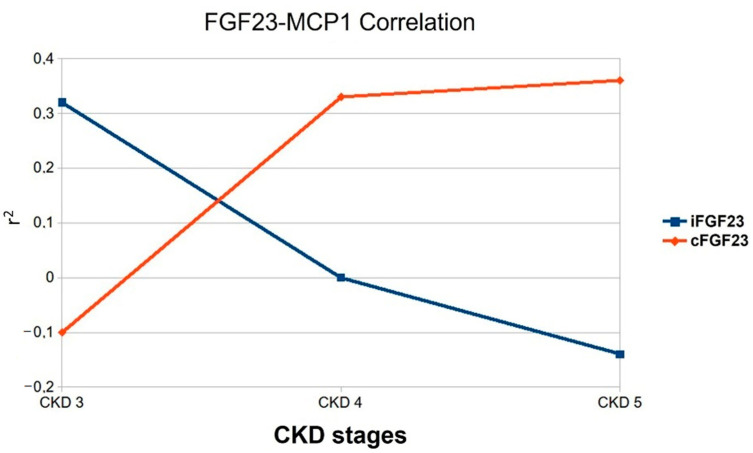
Correlation between MCP1 and i/cFGF23 during CKD stages 3, 4, and 5. Two-tailed Spearman bivariate analysis. *n* = 19 in CKD3, *n* = 25 in CKD4, *n* = 12 in CKD5.

**Table 1 jcm-11-07099-t001:** Demographic and biochemical data.

	TOTAL (56 Patients)	CKD 3 (19 Patients)	CKD 4 (25 Patients)	CKD 5 ND (12 Patients)	*p*-Value 3 vs. 4	*p*-Value 3 vs. 5	*p*-Value 4 vs. 5
Demographic data							
Age (years)	78 ± 8	74 ± 20	81 ± 6	78 ± 8	0.08	0.49	0.12
Gender (m/f)	37/19	15/4	12/13	10/2	0.03	0.76	0.04
Clinical data							
Diabetic subjects *n* (%)	33 (58)	11 (57)	16 (64)	6 (50)	0.68	0.66	0.41
CV events *n* (%)	31 (55)	8 (42)	17 (68)	6 (50)	0.08	0.66	0.29
Tot. cholesterol (mg/dL)	166 ± 37	169 ± 43	160 ± 27	171 ± 47	0.40	0.84	0.31
HDL (mg/dL)	52 ±15	52 ± 13	52 ± 15	55 ± 17	0.98	0.59	0.60
LDL (mg/dL)	88 ± 33	90 ± 38	84 ± 21	92 ± 45	0.51	0.80	0.37
Na (mmol/24 h)	141 ± 2	141 ± 2	141 ± 2.5	141 ± 1	0.67	0.7	0.97
K (mmol/L)	4.6 ± 0.4	4.4 ± 0.4	4.6 ± 0.4	4.9 ± 0.5	0.19	0.07	0.22
Biomarker data							
Ca (mmol/L)	9.2 ± 0.4	9.3 ± 0.4	9.3 ± 0.4	9.0 ± 0.6	0.82	0.12	0.15
Vit. D (ng/mL)	30 ± 15	35 ± 15	29 ± 16	23 ± 13	0.24	0.04	0.32
Phosphoremia (mg/dL)	3.4 ± 0.5	3.1 ± 0.3	3.4 ± 0.5	3.8 ± 0.7	0.12	0.05	0.08
Phosphaturia (mg/24 h)	501 ± 195	571 ± 194	487 ± 199	436 ± 176	0.18	0.04	0.4
PTH (pg/mL)	69 ± 40	54 ± 31	60 ± 30	112 ± 45	0.53	0.0002	0.0002
eGFR	25.5 ± 10,3	37.4 ± 6.5	22.36 ± 3.0	13.1 ± 2.7	<0.0001	<0.0001	0.0003
MCP1 (pg/mL)	465.1± 159.4	384.8 ± 161	478.4 ± 124	552.1 ± 177	0.04	0.15	0.01
FGF23 Cter (RU/mL)	187.7 ± 151.0	151.6 ± 132.0	172.5 ± 99.7	275.2 ± 226.9	0.55	0.06	0.06
FGF23 Intact (pg/mL)	135.0 ± 124.8	78.9 ± 58.9	127.6 ± 70.9	238.7 ± 207.5	0.02	0.02	0.003

*p*-values regarding gender, diabetics, and CV events were calculated with chi-square test; *p* regarding all other parameters were calculated with *t*-test. CV = cardiovascular events, HDL = high-density lipoprotein cholesterol, LDL = low-density lipoprotein cholesterol, Na = sodium, K = potassium, Ca = calcium, PTH = parathyroid hormone, eGFR = estimated glomerular filtration rate, MCP1 = monocyte chemoattractant protein 1, FGF23 Cter = fibroblast growth factor 23 C- terminal, FGF23 Intact = fibroblast growth factor 23 intact. Data are expressed as mean ± standard deviation.

**Table 2 jcm-11-07099-t002:** PUFA profile among CKD stages.

	CKD			*p*-Value		
	Stage 3	Stage 4	Stage 5	3 vs. 4	3 vs. 5	4 vs. 5
PUFA	36.75 ± 3.83	35.63 ± 3.71	39.44 ± 4.27	0.33	0.07	0.009
PUFA n-3	3.59 ± 1.32	2.91 ± 0.65	3.14 ± 1.1	0.03	0.33	0.43
α-Linolenic acid	0.36 ± 0.17	0.3 ± 0.08	0.36 ± 0.15	0.15	0.93	0.174
EPA	0.88 ± 0.62	0.6 ± 0.24	0.58 ± 0.39	0.09	0.13	0.85
DPA	0.42 ± 0.14	0.36 ± 0.1	0.37 ± 0.09	0.04	0.34	0.62
DHA	1.93 ± 0.67	1.65 ± 0.39	1.83 ± 0.66	0.09	0.69	0.30
PUFA n-6	32.93 ± 3.64	32.42 ± 3.62	36.11 ± 4.27	0.64	0.03	0.01
Linoleic acid	22.96 ± 3.77	22.28 ± 3.72	26.13 ± 3.89	0.55	0.03	0.006
Υ-Linolenic acid	0.36 ± 0.17	0.30 ± 0.18	0.36 ± 0.15	0.94	0.16	0.13
DGLA	1.7 ± 0.31	1.84 ± 0.49	1.81 ± 0.36	0.28	0.38	0.84
Arachidonic acid	7.41 ± 1.76	7.37 ± 1.55	7.39 ± 2.16	0.93	0.98	0.96
Osbond acid	0.17 ± 0.07	0.20 ± 0.08	0.17 ± 0.05	0.20	0.89	0.19

The PUFA are expressed as relative % of total considered FA. PUFA: polyunsaturated fatty acids, EPA: eicosapentaenoic acid, DPA: docosapentaenoic acid, DHA: docosahexaenoic acid, DGLA: dihomo-γ-linolenic acid. The values in red and green indicate that these markers decrease and increase in the advanced CKD stage. *p*-value was calculated with t-test. *n* = 19 in CKD3, *n* = 25 in CKD4, *n* = 12 in CKD5).

**Table 3 jcm-11-07099-t003:** Correlation between PUFA and MCP1 among CKD stages. *p* < 0.05, “−” indicates a negative correlation. Two-tailed Spearman bivariate analysis.

MCP1 (pg/mL)	CKD3		CKD4		CKD5	
	r^2^	*p*-value	r^2^	*p*-value	r^2^	*p*-value
PUFA	0.422	0.092	0.147	0.493	−0.231	0.471
n-3	0.162	0.535	−0.032	0.881	−0.28	0.379
18:3n3	−0.209	0.421	−0.256	0.228	0.21	0.513
20:5n3	0.13	0.619	−0.012	0.955	0.119	0.713
22:5n3	−0.02	0.94	0.325	0.122	0.056	0.863
22:6n3	0.223	0.39	−0.056	0.796	−0.326	0.301
n-6	0.377	0.135	0.12	0.576	−0.182	0.572
18:2n6	0.27	0.295	−0.134	0.533	−0.077	0.812
18:3n6	−0.267	0.299	0.187	0.381	0.677	0.016
20:3n6	−0.304	0.236	0.594	0.002	0.371	0.236
20:4n6	0.377	0.135	0.424	0.039	0.007	0.983
22.4n6	−0.347	0.173	0.344	0.099	0.092	0.776
22:5n6	0.062	0.814	0.521	0.009	0.158	0.624

The values in green indicate that this marker increased in the advanced CKD stage. “−” indicates a negative correlation. *n* = 19 in CKD3, *n* = 25 in CKD4, *n* = 12 in CKD5.

## Data Availability

The data presented in this study are available in the insert article and Appendix A.

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
