# Peer review of "MCP1 Could Mediate FGF23 and Omega 6/Omega 3 Correlation Inversion in CKD"

_jcm, 2022, doi:10.3390/jcm11237099_

Round 1
Reviewer 1 Report
I have the opportunity to review this interesting manuscript about the influence of MCP1 in FGF23 and Omega 6/Omega 3 in CKD patients.
I found this study of interest; however, It is a good writing manuscript that may need small typesetting corrections.
I recommend authors address these questions to improve their statements:
1.- Please include the MCP1 statement in the introduction section.
2.- You must double-check the groups according to CKD stages it is a little bit confusing for readers.
3.- I found the first paragraph of the discussion section not necessary.
4.- The final sentence of lines 250-251 can be deleted (at the end of the paragraph)
5.- Before the conclusion section, several small paragraphs could be merged for better reading.
6.- Good luck to the authors in the process
Author Response
Reviewer #1
Submission no: jcm-2021507
Submission title: MCP1 could mediate FGF23 and Omega 6/Omega 3 correlation inversion in CKD
The authors wish to thank the Reviewer for his expert, qualified, and very helpful comments and suggestions which have greatly improved our manuscript. Below is a point-by-point reply to the referees’ queries. We believe we have addressed all the referee’s comments and therefore the manuscript is now suitable for publication.
I have the opportunity to review this interesting manuscript about the influence of MCP1 in FGF23 and Omega 6/Omega 3 in CKD patients. I found this study of interest; however, It is a good writing manuscript that may need small typesetting corrections. I recommend authors address these questions to improve their statements:
1.- Please include the MCP1 statement in the introduction section.
We thank the reviewer for bringing up the issue to our attention, and we have now provided to add the MCP1 statement in the introduction section.
2.- You must double-check the groups according to CKD stages it is a little bit confusing for readers.
The reviewer is entirely correct; in section 2.1 on patient information, a breakdown of patients is mentioned after stating that patients were separated into 3 groups according to their CKD stage, making interpretation a little confusing. We apologize for the mistake. We have removed the phrase " Patients were divided into three subgroups according to their CKD stage".
3.- I found the first paragraph of the discussion section not necessary
According to the reviewer, the paragraph has been deleted, and the discussion has been thoroughly reviewed.
4.- The final sentence of lines 250-251 can be deleted (at the end of the paragraph)
The sentence has been deleted.
5.- Before the conclusion section, several small paragraphs could be merged for better reading
We rewrote the discussion section trying to combine concepts and make reading more fluent.
6.- Good luck to the authors in the process
We thank the reviewer for his careful review, appreciation, and positive feedback on our manuscript.

Reviewer 2 Report
1. the purpose of sections from lines 62-96 are not very clear. as presented, it reads like dumping of information. the real introduction ends on lines 58-61.
2. move ethical approval into patients subsection.
3. define what is HClMe3N, refrain for using acronyms with out definition.
4. statistical section is very barebones, it will be hard for other researchers to replicate the work in future.
5. there is no information on statistical tests applied to check the demographic and clinical variables.
6. clearly mark the biomarker section from the clinican and demographic section.
7. figure1 legend should be more descriptive, provide number of individual in each group. what should the readers understand while looking at this figure.
8. table 2, use a,b,c as superscript on the p values presented. instead of the current way. What was the total fatty acid content for each of these group. this information should be presented in the main text. significant p value cutoff is 0.05, on line 167 how can authors say that p-value of 0.09 is significant. at the most this is indicative of a trend towards higher PUFA. replace "p-value" to "p=" or "p<". use significant instead of significative.
9. pl remove the flowchart of the PUFA biosynthesis. it is not result. remove the box around the figure. there is no box around figure 1.
10. discussion on lines 264-292, do not reads like a discussion but collection of statemtents. much of this area sounds like hypothesis. rewrite the importance of your results in light of the published literature.
11. missing PUFA profile from the urine, how do author explain this to the reader.
Author Response
Reviewer #2
Submission no: jcm-2021507
Submission title: MCP1 could mediate FGF23 and Omega 6/Omega 3 correlation inversion in CKD
The authors wish to thank the Reviewer for his expert, qualified, and very helpful comments and suggestions which have greatly improved our manuscript. Below is a point-by-point reply to the referees’ queries. We believe we have addressed all the referee’s comments and therefore the manuscript is now suitable for publication.
1. the purpose of sections from lines 62-96 are not very clear. as presented, it reads like dumping of information. the real introduction ends on lines 58-61
We apologize for the lack of clarity in the section. We rewrote the paragraph by merging the concepts and adding reliable information and references to the missing MCP1 in the initial version.
2. move ethical approval into patient’s subsection
Following the reviewer’s request, the ethical approval has now been included in the subsection of the patient.
3. define what is HClMe3N, refrain for using acronyms without definition
The reviewer is entirely correct; for better reproducibility of the results, the detail on the acronyms is now reported. The combination of different expertise often leads to underestimating sectorial acronyms, but effectively the clarity of the methods is fundamental and precious. The sentence has been amended starting from the full definition of the chemical formula in the object" hydrochloric acid solution 3N in methanol (HClMe 3N)". In addition, the acronym reported in the tables is now well-defined in the revised version legend.
4. statistical section is very barebones; it will be hard for other researchers to replicate the work in future
Following the reviewer's suggestion, we have rewritten the statistical paragraph.
5. there is no information on statistical tests applied to check the demographic and clinical variables
We have now included the information on statistical tests in the statistical paragraph
6. clearly mark the biomarker section from the clinician and demographic section
The table concerning demographic data has been completely re-edited.
7. figure 1 legend should be more descriptive, provide number of individual in each group. what should the readers understand while looking at this figure.
We agree with the reviewer and have now provided to add the number of individuals of each group in the figure’s legend.
8-
a) table 2, use a,b,c as superscript on the p values presented. instead of the current way.
Thanks for the suggestion; the table has been modified.
b) What was the total fatty acid content for each of these group. this information should be presented in the main text.
Thanks for the suggestion. We added the explanation regarding how fatty acids are represented in method section 2.2, "Fatty acid analysis," as follows: "Both single and fatty acid groups (SFA, MUFA, PUFA, PUFA n-3, PUFA n-6) are expressed as relative percentages of total considered fatty acids, whose value is always 100."
c. significant p value cut-off is 0.05, on line 167 how can authors say that p-value of 0.09 is significant. at the most this is indicative of a trend towards higher PUFA.
Thank you for pointing this out; the real value is p-value 0.009; we apologize for the typing error; now appropriately corrected.
d. replace "p-value" to "p=" or "p<". use significant instead of significative.
Thanks for the note; we have fixed the issues in the main text.
9. pl remove the flowchart of the PUFA biosynthesis. it is not result. remove the box around the figure. there is no box around figure 1.
The flow diagram of the PUFA biosynthesis and the frame around the figure has been deleted.
10. discussion on lines 264-292, do not reads like a discussion but collection of statemtents. much of this area sounds like hypothesis. rewrite the importance of your results in light of the published literature.
We thank the referee for pointing out this. We provide to rewrite the discussion section merging the concepts for better reading.
11. missing PUFA profile from the urine, how do author explain this to the reader.
The reviewer raised a critical point. The urinary fatty acids were not analyzed since the excreted one cannot influence the pathways related to FGF23 and MCP1 in the blood. Fatty acids in urine derive from lost albumin because of a high eGFR, without any biological function except to be considered as a possible biomarker. The focus of the present work was different based on the evaluation of a possible interplay between inflammation FGF23 and fatty acid, and this interaction cannot be found in the ratio of fatty acids. So, urine fatty acid analysis, if helpful to add data regarding the possible role of FA as a kidney function biomarker, doesn't fit with the focus and biological aspect of the present work.

Reviewer 3 Report
It is an interesting work that contains some aspects of improvement:
Patients: as mentioned in the limitations of the study, the number of patients included is very low, especially considering the divisions that are made
Line 105: a subdivision of patients is mentioned after stating that the patients have been separated into 3 groups based on their CKD stage. This is interpreted as a second division of these patients, but it really is not.
It is mentioned that blood and urine samples are collected but it is not explained how they are treated afterwards. In the different analyses, both plasma and serum are used. It would be convenient to mention the treatment after the collection of each sample.
ELISA Kis indicate the commercial that manufactures it but do not mention the reference of the kit used (this would be interesting when replicating these results by other researchers)
Table 1.: both diabetics and cardiovascular events are put in absolute number, this is not the most recommendable for the later statistical treatment since the total n of each group is different, to be able to make comparisons between groups the most recommendable thing would be to show the data What %
Figure 1. eGFR CKD5 is not significant?
Both in this figure and in the rest it would be convenient to indicate the p value against what is being compared
The discussion focuses too much on repeating the information obtained in the results, it would be convenient to discuss in greater depth the interest of these discoveries as well as the possible clinical applicability.
Author Response
Reviewer # 3
Submission no: jcm-2021507
Submission title: MCP1 could mediate FGF23 and Omega 6/Omega 3 correlation inversion in CKD
The authors wish to thank the Reviewer for his expert, qualified, and very helpful comments and suggestions which have greatly improved our manuscript. Below is a point-by-point reply to the referees’ queries. We believe we have addressed all the referee’s comments and therefore the manuscript is now suitable for publication.
It is an interesting work that contains some aspects of improvement:
1. Patients: as mentioned in the limitations of the study, the number of patients included is very low, especially considering the divisions that are made.
We agree with the reviewer and report effectively that this is a limitation, but we think this is a preliminary study containing the important issue. Indeed, following the extensive literature, the significant studies on fatty acid focus on the influence of PUFA intake and /or supplementation on kidney function and consider the CKD cohort without subdividing the several stages. In the present work, we highlight that each CKD stage has distinctive features in terms of FGF23, inflammation that, if underestimated, may lead to a lack of significance and misinterpretation with loss of actual results. Our study is preliminary, but our results, although on a few patients, stimulate further investigation and suggest different issues.
2. Line 105: a subdivision of patients is mentioned after stating that the patients have been separated into 3 groups based on their CKD stage. This is interpreted as a second division of these patients, but it really is not.
The reviewer is entirely correct. The double sentence confuses the concept. We apologize for the mistake. We have removed the phrase " Patients were divided into three subgroups according to their CKD stage".
3. It is mentioned that blood and urine samples are collected but it is not explained how they are treated afterwards. In the different analyses, both plasma and serum are used. It would be convenient to mention the treatment after the collection of each sample.
The reviewer is entirely correct. In the patient's section, we have reported the following sentence: "In addition, the 24h urine samples were collected for routine analysis (eGFR and phosphaturia)".
4. ELISA Kis indicate the commercial that manufactures it but do not mention the reference of the kit used (this would be interesting when replicating these results by other researchers).
Following the reviewer, we have added references in the method section.
5. Table 1.: both diabetics and cardiovascular events are put in absolute number, this is not the most recommendable for the later statistical treatment since the total n of each group is different, to be able to make comparisons between groups the most recommendable thing would be to show the data What %
The new table is now re-edited as follows, reporting the %.
Figure 1. eGFR CKD5 is not significant? Both in this figure and in the rest, it would be convenient to indicate the p value against what is being compared
We appreciate the reviewer's suggestion on the table, and all the information about the p-value among the group is now added in both tables 1 and 2.
6. The discussion focuses too much on repeating the information obtained in the results, it would be convenient to discuss in greater depth the interest of these discoveries as well as the possible clinical applicability.
In accordance with the reviewer, we rewrote the section trying to combine concepts and discuss them better, including some possible clinical strategies, and hoping to hope to have made reading more fluent.

Round 2
Reviewer 2 Report
The authors has addressed all the queries to my satisfaction.